# A brief measure of complete subjective well-being in Germany: A population-based validation of a German version of the Flourish Index (FI) and the Secure Flourish Index (SFI)

Sebastian Sattler[1,2,3]*, Renae Wilkinson[4], Matthew T. Lee[4,5]

1 Faculty of Sociology, Bielefeld University, Bielefeld, Germany, 2 Institute of Sociology and Social Psychology, University of Cologne, Cologne, Germany, 3 Pragmatic Health Ethics Research Unit, Institut de Recherches Cliniques de Montréal, Quebec, Canada, 4 Human Flourishing Program, Institute for Quantitative Social Science, Harvard University, Cambridge, CA, United States of America, 5 Institute for Studies of Religion, Baylor University, Waco, TX, United States of America

* sebastian.sattler@uni-bielefeld.de

**Data Availability Statement:** The data are now also archived in our institutional repository: https://doi.org/10.4119/unibi/2978689.

## Abstract

Measuring subjective well-being in a multidimensional, valid, reliable, and parsimonious way is important for both social science research and social policy. Here, we present an efficient measure of distinct domains of subjective well-being and overall flourishing. The Flourishing Index (FI) consists of five sub-domains: 1. happiness and life satisfaction, 2. physical and mental health, 3. meaning and purpose, 4. character and virtue, and 5. close social relationships. The Secure Flourishing Index (SFI) adds the sub-domain financial and material stability, which is thought to be necessary to sustain the other domains over time. We developed a German version of these measures in a multi-stage translation and scale testing process. The results of an exploratory factor analysis in Study 1 ($N = 192$) suggest a unidimensional structure of the FI and a two-dimensional structure of the SFI. Moreover, both indices (and most sub-domains) revealed acceptable to good reliability. The factor structures were confirmed in Study 2 ($N = 13,268$). We provide indications for measurement invariance of both indices with regard to gender and age. We furthermore examined inter-correlations with related constructs such as importance of health, self-efficacy, and social support. Study 3 ($N = 317$) finds evidence for high convergent validity of both the FI and the SFI with overall well-being as well as sub-scores of the PERMA-Profiler. These results suggest that the FI and the SFI are efficient measures of distinct domains of subjective well-being and overall flourishing. Our translation of the FI and SFI, along with the empirical relationships that we found among the measures that we reviewed, will help scholars in Germany (and beyond) explore an expanded range of domains of well-being, including the comparatively neglected domains of character and virtue, physical health, and financial and material stability.

**Funding:** This work was supported by the German Research Foundation [grant number: SA 2992/2-1; to S.S.]. Additional support was provided by the John Templeton Foundation [grant number: 62022; to M.T.L.], the Kern Family Foundation [grant number: 2019-01467; to M.T.L. and R.W.], the Robert Wood Johnson Foundation [grant number: 74275; to M.T.L.], and Aetna, Inc [grant number: A33796; to M.T.L.]. The funders had no role in study design, data collection and analysis, decision to publish, or preparation of the manuscript.

**Competing interests:** The authors have declared that no competing interests exist.

## Introduction

Well-being has been a long-standing concern in the social sciences and humanities [1–3]. And in recent years, research in this interdisciplinary field has expanded dramatically [4, 5]. There is no consensus regarding the essential elements of well-being, but the US Centers for Disease Control and Prevention has indicated that it would minimally include a positive life evaluation, a sense of positive affect, and the experience of good physical health [4, see also 6 on the relationship between well-being and flourishing]. Investigating well-being of populations is important as an end in itself because people almost universally desire to do or be well across multiple domains [7]. Well-being is also associated with reduced risk of chronic disease and mortality [8], which are indicators of population health that are of primary interest to epidemiologists and health practitioners. Additionally, research on well-being helps us to understand the impacts of social, psychological, and socioeconomic variables on the functioning of society, permitting the identification of social policy "blind spots" that shape outcomes that are consequential for all people, including social dislocations like the Arab uprisings and Brexit [9]. Rigorous empirical data on what makes people happy and well off, along with the ability to assess change in well-being over time and examine how policy decisions may affect well-being, permit the identification of sub-groups with low well-being in order to develop targeted interventions [10]. Next to information such as GDP or life expectancies, measuring and monitoring well-being is important for governments and decision-makers in organizations as well as the general public [11].

According to the latest World Happiness Reports, 75% of the differences in life evaluations (e.g., happiness and life satisfaction) across countries is explained by six factors. Four of these variables reflect the social fabric of a society (i.e., having someone to count on in times of trouble, as sense of freedom to make life choices, generosity, and a trustworthy environment). Moving countries with the lowest values on these four variables up to the world average would raise life evaluations dramatically. This increase would be more than that associated with similar changes in both income per capita and healthy life expectancy [12]. Therefore, measures of well-being need to be multi-dimensional to cover the most important facets of this construct, including aspects beyond GDP and physical health, but they also must be reliable and valid in order to be used in representative population samples [13]. This is why narrow constructs such as "health" or "wellness" have given way to more holistic appraisals of "complete well-being," or "flourishing" [7, 11, 14]. As measures of flourishing—and related constructs such as well-being—have proliferated, the field has produced findings that are at times inconsistent across studies and conceptualizations, and difficult to reconcile [11]. Some psychological measures of "flourishing" such as Seligman's PERMA might be reducible to "subjective well-being" rather than the more complete appraisal suggested by the term [5]. More generally, many measures of well-being are highly correlated but not identical, leading to the conclusion that the "choice of well-being measure should reflect [the researcher's] theoretical aims" [15]. Suggestions have been offered about how to encourage this progress [11, 16], although there are dissenting voices in the dialogue about the path forward [17].

At its core, flourishing connects with terms such as growth, prosperity, and thriving [14]. Determining whether a person is flourishing involves a holistic appraisal of the most important ends of human life, including but also going beyond "foundational concerns" such as positive affect, economic prosperity, or physical health [18]. This assessment might involve both objective and subjective indicators, as well as an appraisal of community conditions [14]. However, it is sometimes only possible to collect a small number of subjective indicators and a parsimonious measure of key domains of individual flourishing is therefore desirable [4, 14, 16]. These domains may be valued at somewhat different levels across cultures, but there appears to be a high level of consensus about which domains most people would include [7, 19–21].

Flourishing has been understood from a variety of vantage points, including philosophically, psychologically, and from a cross-disciplinary perspective [22]. The various measures of flourishing all have strengths and limitations [11], but a cross-disciplinary measure that builds on both philosophical and psychological scholarship has advantages, including the incorporation of domains that are often missing from other measures, such as physical health, good character, and financial well-being [22]. Human flourishing, as a state of complete well-being at the individual level, thus entails at least five domains that are recognized as ends in themselves: happiness and life satisfaction, physical and mental health, meaning and purpose, character and virtue, and close social relationships [7, 22]. Some level of financial and material stability is required to sustain the domains over time and might therefore be considered an indispensable sixth domain [7]. Beyond these essential domains, consensus becomes more difficult. For example, those who are religious value communion with God (or the transcendent), whereas the non-religious might not. But it is difficult to imagine a flourishing life that would not include the five fundamental domains and the enabling domain of material stability.

The domains of flourishing are both outcomes shaped by a host of individual and social factors and also predictors of other outcomes. For example, evidence suggests that the Covid-19 pandemic has had important effects on many, but not all, of the domains of flourishing. From January to June of 2020, a national survey suggested that adults in the U.S. experienced significant declines in health, happiness, and financial stability. But importantly, the character and virtue domain did not decline over this same time period [23]. Another study found that the domains of flourishing—including life satisfaction, purpose, and physical and mental health—predicted higher levels of hope: "a disposition towards having an attentional focus on the possibility that the future will be good" [24]. The relationships in these findings were bidirectional, with increased levels of hope also predicting higher levels of health and well-being. Furthermore, research indicates that financial conditions were associated with subsequent self-reported measures of physical and mental health, independent of confounders [25]. Similarly, character has been found to promote a variety of health and well-being outcomes [26].

## The Flourish Index (FI) and the Secure Flourish Index (SFI)

The twelve-item Secure Flourish Index (SFI) is a measure of flourishing that is short enough to be practical in variety of settings, including in the workplace and in population surveys where longer measures are not able to be included, but it is still comprehensive in its assessment of the most essential domains of "complete" well-being as identified by cross-disciplinary scholarship [7, 20, 22]. The ten-item Flourish Index (FI) omits the two financial and material stability questions but is otherwise identical to the SFI. The FI is appropriate when other measures of financial and material stability are included in the same survey or when these "means" rather than "ends" items are not of interest. The wording of items for the SFI and FI are provided in Weziak-Bialowolska et al. [20]. Developed initially in the United States, the SFI has been translated into several languages and deployed in countries such as China, Mexico, Poland, Sri Lanka, and Cambodia [20] and was recently approved for inclusion in several large, longitudinal cohort studies (e.g., the Growing Up Today Study with 27,000 respondents– http://www.gutsweb.org, and the Nurses Health Study with 275,000 respondents– https://nurseshealthstudy.org). In addition, a major research project focused on flourishing throughout the world was launched in October of 2021 and uses the SFI as a primary outcome measure. Known as the Global Flourishing Study [27], this collaboration with Gallup will begin collecting survey data in 2022 on 240,000 individuals using nationally representative samples in 22 countries representing approximately half of the world's population. The survey will be administered once per year for five years and the resulting longitudinal dataset will be

unprecedented in well-being research. Findings will provide benchmarking for other countries where the SFI might be used in the future.

Both the FI and SFI exhibit favorable psychometric properties based on community and workplace samples, including strong validity as measured by correlations with external measures expected to be related to flourishing, high reliability ($\alpha = 0.89$ for FI; $\alpha = 0.86$ for SFI), and a factor analysis which confirmed groupings of items into domains of flourishing [20]. Cross-cultural research with worker samples using the FI and SFI established configural, metric, and partial scalar measurement invariance, suggesting that the measures are culturally universal [21]. Interestingly, in this study U.S. respondents had the lowest scores on most domains of flourishing as measured by the SFI, with the exception of two domains: happiness and life satisfaction and financial and material stability. Respondents from China had the highest scores for health and close social relationships, while character and virtue scores were relatively high in Cambodia. Respondents in Mexico reported the lowest scores in financial and material stability, but also had the greatest meaning and purpose to their lives [21]. Further cross-cultural research is needed with community samples in order to better understand the patterns of overall and domain-specific flourishing that prevail through the world. We, however, do not know of any other study using the FI and SFI in Germany, probably also because of the absence of a validated measure.

## Current study

Therefore, this paper has five major goals: 1. to translate the FI and the SFI to German; 2. examine the dimensional structure of the items exploratorily and confirmatorily with two independent samples; 3. analyze the internal consistency of the sub-domains and the FI and the SFI; 4. explore the relation of the scales with other instruments in the form of convergent and discriminatory validity; and 5. test the measurement invariance to better understand if the same constructs are equivalently measured across gender and age.

## Descriptions of the stages of the translation process

We based our translation of the Human Flourishing Scale on the international guidelines for cross-cultural adaptation of health-related self-report measures [28, 29]. In step 1, questions were translated from English to German by one professional translator (native German speaker) and by one bilingual survey expert from sociology (see Fig 1 for an overview of the process).

In step 2, back-translations were completed by two bilingual native English-speakers of whom one is a professional translator and one was a sociologist with a background in well-being research, broadly defined, as well as specific familiarity with the English version of the SFI and the social science constructs that it was designed to assess. In step 3, discrepancies between translations and the original items were discussed by two members of the research team (of which one is a German native speaker), two professional linguists in English (one was an English native speaker), one survey expert from sociology (German native speaker), the original two translators, and the two back-translators. The group was informed that the goal is to ensure that the translation is fully comprehensible to a majority of people (e.g., short sentences, as simple as possible, active rather than passive voice; avoiding metaphors) and considers cross-cultural equivalence of source and conceptual equivalence [28]. While some items led to more discussion, e.g., about how to translate and culturally adapt the phrase "purpose of life" or the term "happiness" (D4.2), the entire group arrived at a consensus about the final wording of each item. In step 4, cognitive testing was completed with a divers sample of 18 adult residents in Germany using the think-aloud technique and probing questions [30] to evaluate the comprehensibility and content validity of the items. Probing respondents about their understanding of the meaning item (D4.1), we found that some respondents described it

**Stage 1:** Translation by one professional translator and by one bi-lingual survey expert.

**Stage 2:** Backtranslation by two bilingual native English-speakers of whom one is a professional translator and one with a background in well-being research, broadly defined, as well as specific familiarity with the English version of the SFI and the social science constructs that it was designed to assess.

**Stage 3:** Expert round on the translations with members of the research team, two professional linguists, one survey expert, the original translators, and the two back-translators.

**Stage 4:** Cognitive pretest with a diverse sample of adult residents in Germany ($N$=18).

**Stage 5a:** Quantitative pretest with a sample of residents in Germany ($N$=192) to descriptively examine the items, sub-domains, and the scale as well as to test the dimensional structure with Exploratory Factor Analysis and reliability with Cronbach's α.

**Stage 5b:** Large-scale study with a sample of residents in Germany ($N$=13,268) to descriptively re-examine the items, sub-domains, and scale; to test the dimensional structure with Confirmatory Factor Analysis; to examine Measurement Invariance regarding gender and age; to explore construct validity through examining correlations with related measures (e.g., importance of health, personal burnout); and to re-examine reliability with Cronbach's α and McDonald ω.

**Stage 5c:** Testing the convergent validity with the German version of the PERMA-profiler measuring well-being with a sample of residents in Germany ($N$=317).

**Fig 1. Descriptions of the stages of the translation process and the validation of the scale.**

as demanding to answer questions about their meaning of life and had to think about it, but could finally answer the item. Moreover, we also found that the virtue item (D4.2) was understood as intended (e.g., regarding the ability to delay gratification for better outcomes), however, it made respondents think. Based on these 4 steps, we created a version to be tested quantitatively for their scale properties (Step 5), what will be described in the following.

## Study 1

The first study aimed at examining the translated German items, sub-domains, and the FI as well as the SFI scale descriptively, and it explored the dimensional structure as well as scale reliability.

## Methods

**Participants.** We conducted a quantitative web-based assessment with an offline-recruited nationwide sample (multi-stage random process based on the ADM (Arbeitskreis Deutscher Markt- und Sozialforschungsinstitute e.V.) telephone master sample) consisting of adult respondents in Germany, recruited via the forsa.omninet panel. The gross sample consisted of 264 invited individuals of which 237 (89.8%) individuals provided written consent to participate, and 200 (84.4%) completed the multithemed study between August and September 2020. Only a very small fraction of the respondents has chosen "does not apply" as response option across the FI and SFI items. This number was on average 2.3 out of 203 respondents, with the highest numbers for the virtue item D4.2 ($N = 7$; 3.4%) and the character item D4.1 ($N = 5$; 2,5%). Thus, our further analysis comprises of answers of 192 individuals due to listwise deletion of missing responses (45.8% women, mean age: 45.64, age range: 18 to 80 years). Participants received bonus points as incentives (approx. $2.30, convertible to vouchers, a ticket for a charity lottery or a donation to UNICEF) upon completion. The ethics committee of the University of Erfurt approved the study (reference number: EV-20190917) [31].

**Instruments.** *Flourishing Index (FI).* The FI consists of five dimensions with two questions and statements per dimension (see wording in Table 1). Items were measured on 11-point scales (from 0 to 10) with higher scores indicating higher levels of human flourishing. Respondents could also choose "does not apply" as a response option. To compute the FI, all items were averaged and for domain-specific indices, each pair of items was averaged.

*Secure Flourishing Index (SFI).* The SFI comprises–in addition to the FI–a measure of the availability of material and financial prerequisites to maintain the state of flourishing over time. Again, both items were measured on 11-point scales (from 0 to 10) with higher scores indicating higher levels of human flourishing. Respondents could also choose "does not apply" as a response option. For the sub-domain, both items were averaged and to compute the SFI all items of the sub-domains were averaged.

## Statistical analysis

We provide descriptive information about the distribution of the answers (i.e., mean values, standard deviations, minimum, maximum, skewness, and excess kurtosis) as well as Pearson correlation coefficients between items and sub-domains. To examine the correspondence of the items to the theoretical grouping, we ran exploratory factor analysis (*EFA*) via principal axis factoring. We used oblimin rotation for acknowledging the correlation of domains as in other research on flourishing [21, 32]. We also assessed the adequacy of the data through the Kaiser-Meyer-Olkin (*KMO*) measure and Bartlett's test of sphericity. Moreover, we conducted a reliability analysis (*Cronbach´s α*) for each sub-dimension as well as for the FI and SFI.

## Results

**Descriptive results and correlations.** The average response time for all items was 104.80 seconds (*SD* = 49.54). Table 2 shows the mean, standard deviation, minimum, maximum, skewness, and excess kurtosis for all SFI and FI as well as their sub-dimensions (S1 Table in S1 File, for item-specific information). When testing for the normality of the data, we found that all scores are in acceptable limits (±2 as defined by Field 2009) regarding *skewness$_{max}$* = -1.15 and *kurtosis$_{max}$* = 0.43, while the negative skewness implies more right-handed tails, the latter number points towards a minimally leptokurtic distribution, i.e., having minimally fatter tails. Data show that all sub-domains correlate positively of which especially high correlations exist

**Table 1. Flourish Index (FI) and Secure Flourish Index (SFI)–structure, items and exploratory factor analysis (with oblimin rotation) ($N_{Study\ 1}$ = 192).**

| Measure | Domain | Statement/question and response options* | FI$_{D1-5}$ | SFI$_{D1-6}$ | |
|---|---|---|---|---|---|
| | | | F1 | F1 | F2 |
| FI | D1. Happiness and Life Satisfaction | D1.1 Overall, how satisfied are you with life as a whole these days? *0 = not satisfied at all, 10 = completely satisfied* Wie zufrieden sind Sie gegenwärtig alles in allem mit Ihrem Leben? *0 = überhaupt nicht zufrieden, 10 = voll und ganz zufrieden* | 0.768 | 0.735 | 0.248 |
| FI | D1. Happiness and Life Satisfaction | D1.2 In general, how happy or unhappy do you usually feel? *0 = extremely unhappy, 10 = extremely happy* Wie glücklich oder unglücklich fühlen Sie sich normalerweise im Allgemeinen? *0 = extrem unglücklich, 10 = extrem glücklich* | 0.832 | 0.804 | 0.162 |
| FI | D2. Mental and Physical Health | D2.1 In general, how would you rate your physical health? *0 = poor, 10 = excellent* Wie würden Sie Ihre körperliche Gesundheit im Allgemeinen bewerten? *0 = sehr schlecht, 10 = ausgezeichnet* | 0.635 | 0.582 | 0.256 |
| FI | D2. Mental and Physical Health | D2.2 How would you rate your overall mental health? *0 = poor, 10 = excellent* Wie würden Sie Ihre psychische Gesundheit im Allgemeinen bewerten? *0 = sehr schlecht, 10 = ausgezeichnet* | 0.767 | 0.725 | 0.225 |
| FI | D3. Meaning and Purpose | D3.1 Overall, to what extent do you feel the things you do in your life are worthwhile? *0 = not at all worthwhile, 10 = completely worthwhile* Inwieweit haben Sie das Gefühl, dass das, was Sie in Ihrem Leben tun, einen Nutzen hat? *0 = überhaupt nicht, 10 = voll und ganz* | 0.693 | 0.719 | -0.006 |
| FI | D3. Meaning and Purpose | D3.2 I understand my purpose in life. *0 = strongly disagree, 10 = strongly agree* Ich weiß, was der Sinn meines Lebens ist. *0 = stimme überhaupt nicht zu, 10 = stimme voll und ganz zu* | 0.744 | 0.748 | 0.099 |
| FI | D4. Character and Virtue | D4.1 I always act to promote good in all circumstances, even in difficult and challenging situations. *0 = not true of me, 10 = completely true of me* Ich setze mich immer für das Gute ein, selbst in schwierigen und herausfordernden Situationen. *0 = trifft nicht auf mich zu, 10 = trifft voll und ganz auf mich zu* | 0.318 | 0.355 | -0.127 |
| FI | D4. Character and Virtue | D4.2** I am always able to give up some happiness now for greater happiness later. *0 = not true of me, 10 = completely true of me* Ich bin immer in der Lage, auf momentane Wünsche zu verzichten, um mir in der Zukunft größere Wünsche zu erfüllen. *0 = trifft nicht auf mich zu, 10 = trifft voll und ganz auf mich zu* | 0.304 | 0.294 | 0.109 |
| FI | D5. Close Social Relationships | D5.1 I am content with my friendships and relationships. *0 = strongly disagree, 10 = strongly agree* Ich bin mit meinen Freundschaften und Beziehungen zufrieden. *0 = stimme überhaupt nicht zu, 10 = stimme voll und ganz zu* | 0.669 | 0.691 | 0.064 |
| FI | D5. Close Social Relationships | D5.2 My relationships are as satisfying as I would want them to be. *0 = strongly disagree, 10 = strongly agree* Meine Beziehungen sind so erfüllend, wie ich es mir wünsche. *0 = stimme überhaupt nicht zu, 10 = stimme voll und ganz zu* | 0.736 | 0.760 | 0.044 |
| SFI | D6. Financial and Material Stability | D6.1 How often do you worry about being able to meet normal monthly living expenses? *0 = worry all of the time, 10 = do not ever worry* Wie oft machen Sie sich Sorgen, ob Sie Ihre monatlichen Lebenskosten decken können? *0 = sorge mich ständig, 10 = sorge mich nie* | — | 0.341 | 0.856 |
| SFI | D6. Financial and Material Stability | D6.2 How often do you worry about safety, food, or housing? *0 = worry all of the time, 10 = do not ever worry* Wie oft machen Sie sich Sorgen um Ihre persönliche Sicherheit, die Möglichkeit, sich Lebensmittel leisten zu können, oder Ihre Wohnsituation? *0 = sorge mich ständig, 10 = sorge mich nie* | — | 0.351 | 0.854 |

**Notes**: The original instruction was: „Please respond to the following questions on a scale from 0 to 10:". The translated instruction was: „Bitte beantworten Sie die folgenden Fragen auf einer Skala von 0 bis 10:". *Respondents could also choose "does not apply" as response option. **Above, we display the wording of Study 2, the wording in Study 1 was "Ich bin immer in der Lage, auf momentane Freude zu verzichten, um in der Zukunft größere Freude zu erlangen." Bartlett's test of sphericity: $Chi^2_{FI}$ = 960.11; $p_{FI}$<0.001 and $Chi^2_{SFI}$ = 1336.77, $p_{SFI}$<0.001; Kaiser-Meyer Olkin (*KMO*) = 0.87 (FI) and 0.85 (SFI)

**Table 2. Flourish Index (FI) and Secure Flourish Index (SFI)–correlations, descriptive statistics, and reliability ($N_{Study\ 1}$ = 192).**

| Measure | Domain | Pearson´s correlation coefficients | | | | | | | | Descriptives | | | | | | Cronbach's α |
|---------|--------|------|------|------|------|------|------|------------|-------------|------|------|------|------|------|------|------|
| | | D1 | D2 | D3 | D4 | D5 | D6 | $FI_{D1-5}$ | $SFI_{D1-6}$ | M | SD | Min | Max | Skew | Kurt | α |
| FI, SFI | D1. Happiness and Life Satisfaction | 1.000 | | | | | | | | 7.13 | 1.75 | 2.00 | 10.00 | -0.81 | -0.04 | 0.81 |
| FI, SFI | D2. Mental and Physical Health | 0.716*** | 1.000 | | | | | | | 7.02 | 1.82 | 1.50 | 10.00 | -0.69 | 0.11 | 0.75 |
| FI, SFI | D3. Meaning and Purpose | 0.630*** | 0.562*** | 1.000 | | | | | | 6.69 | 2.31 | 0.00 | 10.00 | -0.74 | 0.18 | 0.80 |
| FI, SFI | D4. Character and Virtue | 0.293*** | 0.262*** | 0.430*** | 1.000 | | | | | 6.65 | 1.59 | 2.00 | 10.00 | -0.34 | -0.02 | 0.40 |
| FI, SFI | D5. Close Social Relationships | 0.640*** | 0.451*** | 0.606*** | 0.299*** | 1.000 | | | | 6.99 | 2.25 | 0.00 | 10.00 | -0.77 | 0.05 | 0.86 |
| SFI | D6. Financial and Material Stability | 0.494*** | 0.496*** | 0.326*** | 0.152* | 0.329*** | 1.000 | | | 7.50 | 2.79 | 0.00 | 10.00 | -1.15 | 0.20 | 0.94 |
| FI | D1-D5 | 0.848*** | 0.767*** | 0.857*** | 0.562*** | 0.802*** | 0.463*** | 1.000 | | 6.89 | 1.51 | 1.40 | 9.60 | -0.66 | 0.34 | 0.88 |
| SFI | D1-D6 | 0.848*** | 0.782*** | 0.804*** | 0.509*** | 0.760*** | 0.684*** | 0.963*** | 1.000 | 6.99 | 1.53 | 1.50 | 9.67 | -0.77 | 0.43 | 0.88 |

**Notes**: *$p<0.05$

**$p<0.01$

***$p<0.001$; *M* = Mean; *SD* = Standard deviation; *Min* = Minimum; *Max* = Maximum; *Skew* = Skewness; *Kurt* = Excess kurtosis.

between the sub-dimensions D1-Happiness and Life Satisfaction correlates with D2-Mental and Physical Health ($r$ = 0.716), D3-Meaning and Purpose ($r$ = 0.630), and D5-Close Social Relationships ($r$ = 0.640) as well as D3-Meaning and Purpose with and D5-Close Social Relationships ($r$ = 0.606).

**Dimensional structure: Exploratory factor analysis.** A Bartlett's test of sphericity ($Chi^2_{FI}$ = 960.11, $p_{FI}<0.001$; $Chi^2_{SFI}$ = 1336.77, $p_{SFI}<0.001$) indicates that our data are suitable for the planned data reduction strategy, also the *KMO* of 0.87 (FI) and 0.85 (SFI) indicate a high suitability of the data for structure detection. Therefore, we ran *EFA* with oblimin rotation which indicates a one-factor solution for the FI with an Eigenvalue of 4.49 that explains 87.5% of the overall variance. Factor loadings range from 0.304 to 0.832 (Table 1). A second analysis indicates a two-factor solution for the SFI with Eigenvalues of 4.63 and 1.71 that explain 67.5% and 24.9% of the overall variance. Factor loadings for the first factor range from 0.294 to 0.804, while both items of D4-Character and Virtue had relatively low loadings. Both items of the sub-dimensions D6-Financial and Material Stability had loadings of 0.341 (D6.1) and 0.351 (D6.2) on the first factor, but they had most substantial loadings on the second factor. Items of D1 to D5 had no substantial double-loadings on the second factor.

**Reliability analysis.** The reliability of the FI and SFI as well as all sub-domains was determined by means of consistency analysis (*Cronbach's α*). Most *α*-coefficients indicate acceptable to excellent reliability (Table 2) between 0.75 and 0.94. The only exception is D4-Character and Virtue with *α* = 0.40.

## Discussion

The main purpose of this first study was to descriptively examine the items of the FI and SFI that have been translated to German and to explore that dimensional structure as well as the reliability. Our data from an offline-recruited sample of adult residents in Germany found low numbers of "does not apply" indicating that despite the philosophical nature of some items, respondents could easily answer the questions. However, both items from the sub-domain

D4-Character and Virtue had the highest share of "does not apply". Concerning skewness and kurtosis analysis, data reveal no deviation from normality, while answers had minimally right-handed tails and some had leptokurtic distributions. We found a one-factor solution for the FI items with only one item (D4.2) not meeting the common minimum loading of 0.32 [33]. When factor analyzing the SFI by adding the sub-domain D6-Financial and Material Stability, we found a two-factor solution. Again, item D4.2 did not reach the minimum loading. Low or no substantial loadings of this item have been also reported previously with other samples [20, 21]. While both added items constitute the second factor with high loadings, they also have crossloadings (>0.32) on the first factor. Moreover, we found mainly high reliability of the sub-domains and almost excellent reliability for FI and SFI. D4-items, however, only had a poor reliability. Based on the results of this first quantitative testing and given the discussions concerning this item in the previous stages, we refined item D4.2 by replacing the word "Freude" with the word "Wünsche" and adapted the rest of the item accordingly. We made this amendment in the hope that the item is now easier to understand.

## Study 2

The second study aims at replicating the findings of Study 1 with confirmatory analysis of the dimensional structure and measurement accuracy of the measures, especially after refining item D4.2. We also tested measurement invariance across gender and age, because previous research suggested that indicators of flourishing can partially vary along both indicators [e.g., 1]. Internal consistency was re-assessed. Moreover, Study 2 explores the discriminant validity of the scales with subjective (e.g., fear of failure) and objective (e.g., mental illness diagnosis) measures of psychological distress, social cohesion (e.g., perceived social support), personal resources (e.g., self-efficacy), and subjective (e.g., ability to save money) and objective measures of economic well-being (e.g., household equivalence income).

### Methods

**Participants.** For Study 2, respondents were again recruited via the forsa.omninet panel. They belong to the second wave of the ENHANCE study. The initial sample was nationally representative with regard to sex, age, education, and province for adult the adult (18 or older) residents in Germany (having internet access, which applies to about 95% of all households, [34]). For this wave, 24,683 individuals have been invited of which 17,818 (72.2%) individuals provided written consent to participate, and 15,235 (85.5%) completed the multithemed study between October and December 2020. Only a very small fraction of the respondents has chosen "does not apply" as response option across the FI and SFI items (on average 1.2%, with the highest share for D3.2, namely 2.5%). After listwise removal of missing values on any measure our analysis comprises of answers of 13,268 individuals (49.74% women, mean age: 51.79, age range: 18 to 92 years). Participants received bonus points as incentives (approx. $3.00, convertible to vouchers, a ticket for a charity lottery or a donation to UNICEF) upon completion. The ethics committee of the University of Erfurt approved the study (reference number: EV-20200805). The data that support the findings of this study are openly available in the institutional repository of Bielefeld University [31].

**Instruments.** *Flourishing Index (FI) and Secure Flourishing Index (SFI).* The FI and SFI were assessed in wave 2 as in Study 1 with item D4.2 being adapted (Table 1).

*Importance of health.* Respondents indicated the importance of health to them (in wave 2) on a scale from "not important" [0] to "very important" [10] [adapted from 35]. See Table 3 for descriptive statistics and information on Cronbach's α (where applicable) of all validation measures.

**Table 3. Descriptive statistics and reliability of the validation measures ($N_{Study\ 2}$ = 13,268).**

| Measure | M | SD | Min | Max | Cronbach's α |
|---|---|---|---|---|---|
| Importance of health | 10.32 | 1.13 | 1 | 11 | — |
| Mental illnesses diagnosed | 0.30 | 0.46 | 0 | 1 | — |
| Fear of failure | 2.27 | 0.92 | 1 | 5 | 0.82 |
| Berlin Social Support Scale (BSSS) | 3.98 | 0.97 | 1 | 5 | 0.93 |
| General Self-efficacy Beliefs (ASKU) | 4.05 | 0.60 | 1 | 5 | 0.84 |
| Temper | 2.65 | 0.91 | 1 | 5 | 0.65 |
| Materialism | 2.02 | 0.80 | 1 | 5 | 0.54 |
| Household equivalence income | 2485.99 | 9875.30 | 0 | 1.000.000 | — |
| Possibility to save money | 0.85 | 0.36 | 0 | 1 | — |

**Notes:** *M* = Mean; *SD* = Standard deviation; *Min* = Minimum; *Max* = Maximum.

*Mental illnesses diagnosed.* Respondents were asked (in wave 2) several questions about whether they were diagnosed with illnesses such as depression, Attention Deficit Disorder (ADD), Attention Deficit Hyperactivity Disorder (ADHD), phobias, dementia, or other mental illnesses. Responses were coded into a binary indicator for at least one mental illness diagnosis, whereby the value 0 indicates "no mental illness" and 1 "yes, at least 1 mental illness".

*Fear of failure.* The tendency to avoid possible failure was measured (in wave 2) with two items of the German version of the Revised Achievement Motives Scale [36] with answers ranging from "does not apply at all" [1] to "completely applies" [5]. A sample item is "Even if nobody is watching, I feel quite anxious in new situations."

*Berlin Social Support Scale (BSSS).* Instrumental and emotional social support was measured (in wave 2) with four items of the widely used Berlin Social Support Scale [37] with response options "does not apply at all" [1] to "completely applies" [5]. A sample item is: "Whenever I am sad, there are people who cheer me up."

*General Self-Efficacy Beliefs (ASKU).* We used the validated three-item short scale for measuring general self-efficacy beliefs in wave 1 [38] with answers ranging from "does not apply at all" [1] to "completely applies" [5]. A sample item is: "In difficult situations I can rely on my abilities."

*Temper.* Temper was measured with two items of a well-established self-control scale trait scale in wave 1 [39, 40]. Response options ranged from "does not apply" [1] to "completely applies" [5]. A sample item is: "I lose my temper pretty easily."

*Materialism.* We assessed (in wave 1) whether individuals tend to use possessions to judge the success of others and oneself with the German version [41] of the sub-domain success of the trait measure Material Values Scale [42]. Response options ranged from "does not apply" [1] to "completely applies" [5]. A sample item is: "The things I own say a lot about how well I'm doing in life."

*Household equivalence income.* Participants estimated their monthly household net income in an open-ended question (in wave 2), adapted from Beckmann et al. [43]. If no answer was given, income categories were shown. To estimate equivalence income, the number of persons per household was assessed. Equivalence income was then computed with the OECD-modified scale, in which the first adult has a weight of 1, additional adults weigh 0.5 each, and each child (under 14) has a weight of 0.3 [44]. Income was divided by the weight. If participants provided an income category only, the mean value of the category was taken. Since the last category was open-ended, its mean value was estimated [45].

*Possibility to save money.* Respondents were asked (in wave 2) if they usually have a certain amount of money left to save each month, for instance for larger purchases, emergencies, or asset building [46]. Responses options were "no" [0] and "yes" [1].

**Statistical analysis.** To test the priori formulated measurement model based on the *EFA*, we used confirmatory factor analysis [CFA; 47]. Two specifications were tested: five- (for FI) and six- (for SFI) factor models, wherein items were grouped onto their respective domains; and second-order factor models, with items aggregated into composite measures corresponding to human flourishing or secure human flourishing. *CFA* models were estimated using maximum likelihood and the factors were identified via latent variance. Goodness of fit of the *CFA* models was assessed with: the comparative fit index (*CFI*), the root mean square error of approximation (*RMSEA*), and the standardized root mean square residual (*SRMR*). Values of <0.95 for the *CFI* indicate an acceptable fit, while values in the range of 0.00–0.08 for the *RMSEA* and *SRMR* are considered satisfactory [48].

In tests of measurement invariance of the FI and SFI across gender and age groupings (45 and below, 46–60, and 61 and above), changes in fit indices were examined. We used multi-group CFA to test successively for configural, metric and scalar measurement invariance. For the configural model, the factor loadings, intercepts, and residual variances were free across groups and the factor means were fixed at zero in all groups. For the metric model, the factor loadings were constrained to be equal across groups, intercepts and residual variances were free across groups, and factor means were set at zero in all groups. For the scalar mode, the factor loadings and intercepts were constrained to be equal across groups, residual variances were free across groups, and the factor means were fixed at zero in one group and free in the other groups. Changes in *CFI* less than -0.01 and in *RMSEA* less than 0.015 were utilized as criteria for non-invariance [49].

Reliability (measurement accuracy) was assessed with *Cronbach's α* and the *Omega-coefficient (ω)* indicating how well the latent variable reflects the common variance of all items [50]. We used Pearson correlations to assess discriminant validity.

## Results

**Descriptive results and correlations.** Table 4 shows the mean, standard deviation, minimum, maximum, skewness, and excess kurtosis for all SFI and FI as well as their sub-dimensions (S2 Table in S1 File, for item-specific information). When testing for normality, we found that all *skewness* and *kurtosis* scores of all domains are in acceptable limits, while on the item level, items have right-handed tails and few had a minimally leptokurtic distribution. The mean value of the refined item D4.2 as well as of domain D4. Character and Virtue substantially increased.

Table 5 shows the factor loadings of the *CFA* for the second-order factor models for the FI and SFI, and the results for the FI and SFI are presented in Figs 2 and 3.

The five- (for the FI) and six- (for the SFI) factor models and the second-order factor models for the FI and SFI showed satisfactory fit (Table 6). The excellent fit statistics for the five-factor model (FI) and six-factor model (SFI) empirically confirm both indexes have hierarchical structures; that is, they are comprised of items grouped onto their respective domains. Additionally, the fit statistics for the second-order factor models for the FI and SFI reveal satisfactory fit, indicating the items of the FI and the items of the SFI can be aggregated into composite measures–the Flourish Index and Secure Flourish Index, respectively.

**Measurement invariance testing.** The results of the tests of invariance provide support for the generalizability and robustness of the FI and SFI factor structures for gender (Table 7). The maximum differences between the fit statistics when comparing more restrictive scalar

**Table 4. Flourish Index (FI) and Secure Flourish Index (SFI)–correlations, descriptive statistics, and reliability ($N_{Study\ 2}$ = 13,268).**

| Measure | Domain | Pearson´s correlation coefficients | | | | | | | | Descriptives | | | | | | Cronbach's $\alpha$ |
|---|---|---|---|---|---|---|---|---|---|---|---|---|---|---|---|---|
| | | D1 | D2 | D3 | D4 | D5 | D6 | $FI_{D1-5}$ | $SFI_{D1-6}$ | M | SD | Min | Max | Skew | Kurt | |
| FI, SFI | D1. Happiness and Life Satisfaction | 1.000 | | | | | | | | 6.94 | 1.86 | 0 | 10 | -0.97 | 0.79 | 0.86 |
| FI, SFI | D2. Mental and Physical Health | 0.725*** | 1.000 | | | | | | | 6.76 | 1.94 | 0 | 10 | -0.77 | 0.18 | 0.72 |
| FI, SFI | D3. Meaning and Purpose | 0.632*** | 0.569*** | 1.000 | | | | | | 6.93 | 2.23 | 0 | 10 | -0.86 | 0.27 | 0.80 |
| FI, SFI | D4. Character and Virtue | 0.359*** | 0.340*** | 0.454*** | 1.000 | | | | | 7.23 | 1.64 | 0 | 10 | -0.60 | 0.46 | 0.49 |
| FI, SFI | D5. Close Social Relationships | 0.614*** | 0.477*** | 0.512*** | 0.357*** | 1.000 | | | | 7.08 | 2.32 | 0 | 10 | -0.93 | 0.33 | 0.90 |
| SFI | D6. Financial and Material Stability | 0.390*** | 0.371*** | 0.320*** | 0.235*** | 0.259*** | 1.000 | | | 7.86 | 2.49 | 0 | 10 | -1.33 | 0.95 | 0.91 |
| FI | D1-D5 | 0.857*** | 0.797*** | 0.825*** | 0.616*** | 0.783*** | 0.403*** | 1.000 | | 6.99 | 1.56 | 0 | 10 | -0.73 | 0.38 | 0.89 |
| SFI | D1-D6 | 0.841*** | 0.785*** | 0.795*** | 0.592*** | 0.743*** | 0.620*** | 0.968*** | 1.000 | 7.13 | 1.52 | 0 | 10 | -0.78 | 0.51 | 0.88 |

*Notes*: ***$p<0.001$; *M* = Mean; *SD* = Standard deviation; *Min* = Minimum; *Max* = Maximum; *Skew* = Skewness; *Kurt* = Excess kurtosis.

models to less restrictive metric models were 0.002 for FI and SFI, suggesting the factor loadings for both measures are equivalent for women and men. The results also indicate the item intercepts are equivalent across the gender groups. In relation to age, the maximum changes in fit statistics suggest metric invariance for the SFI and FI factor structures, as well as scalar

**Table 5. Standardized factor loadings for the second-order confirmatory factor analysis for Flourish Index (FI) and Secure Flourish Index (SFI) ($N_{Study\ 2}$ = 13,268).**

| Statement/question/domain | $FI_{D1-5}$ | $SFI_{D1-6}$ |
|---|---|---|
| D1.1 Overall, how satisfied are you with life as a whole these days? | 0.862 | 0.865 |
| D1.2 In general, how happy or unhappy do you usually feel? | 0.888 | 0.885 |
| D2.1 In general, how would you rate your physical health? | 0.638 | 0.640 |
| D2.2 How would you rate your overall mental health? | 0.885 | 0.882 |
| D3.1 Overall, to what extent do you feel the things you do in your life are worthwhile? | 0.873 | 0.874 |
| D3.2 I understand my purpose in life. | 0.776 | 0.775 |
| D4.1 I always act to promote good in all circumstances, even in difficult and challenging situations. | 0.623 | 0.616 |
| D4.2 I am always able to give up some happiness now for greater happiness later. | 0.527 | 0.533 |
| D5.1 I am content with my friendships and relationships. | 0.926 | 0.926 |
| D5.2 My relationships are as satisfying as I would want them to be. | 0.896 | 0.896 |
| D6.1 How often do you worry about being able to meet normal monthly living expenses? | — | 0.899 |
| D6.2 How often do you worry about safety, food, or housing? | — | 0.936 |
| D1. Happiness and life satisfaction | 0.958 | 0.957 |
| D2. Mental and physical health | 0.912 | 0.916 |
| D3. Meaning and purpose | 0.820 | 0.821 |
| D4. Character and virtue | 0.647 | 0.650 |
| D5. Close social relationships | 0.704 | 0.700 |
| D6. Financial and material stability | — | 0.457 |

*Notes*: All alpha-if-item-deleted values indicated FI and SFI would not improve with omission of item/domain.

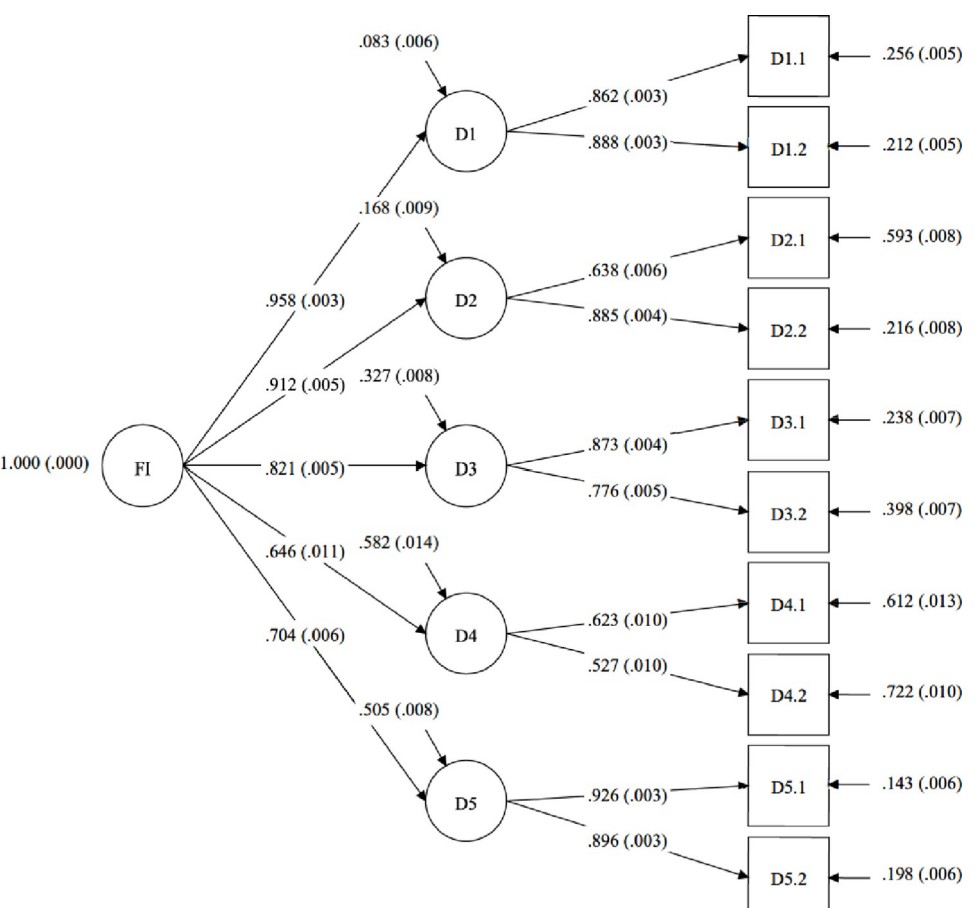

**Fig 2. Standardized factor loadings (with standard errors in brackets) for the second-order confirmatory factor analysis for the Flourish Index (FI)** ($N_{Study\ 2}$ = 13,268).

invariance for the SFI. The change in *CFI* for FI, however, is on the cusp of acceptable when comparing the metric and scalar models ($\Delta CFI$ = -0.01), which could suggest the intercepts of the items are not similar for people of different ages. Still, the results across fit statistics for the most restrictive (scalar) model confirm sufficient fit.

**Reliability analysis.** Reliability measured with *Cronbach´s $\alpha$* indicates acceptable to excellent reliability of the SFI and FI as well as all sub-domains ($\alpha$'s ranged from 0.72 to 0.91, Table 4), while the $\alpha$ from domain D4 increased from 0.40 to 0.49. Measurement accuracy assessed with *Omega* coefficients shows a sufficient degree of scale reliability for both scales ($\omega_{FI}$ = 0.89, $\omega_{SFI}$ = 0.89). With the exception of domain D4, this also applies to the subscales ($\omega_{D1}$ = 0.87, $\omega_{D2}$ = 0.74, $\omega_{D3}$ = 0.81, $\omega_{D4}$ = 0.50, $\omega_{D5}$ = 0.91, and $\omega_{D6}$ = 0.91).

**Discriminant validity.** As expected, individuals diagnosed with a mental illness, with more fears of failure, higher temper, and more materialistic tendencies reported lower FI and SFI values (all *p-values* < 0.001, Table 8). Positive correlations were found with the Berlin Social Support Scale, the importance of health, General Self-Efficacy Beliefs, and the possibility to save money (all *p* < 0.001). Household equivalence income only poorly correlated with FI and SFI. See Table 8 for more details on correlations with the six flourishing domains.

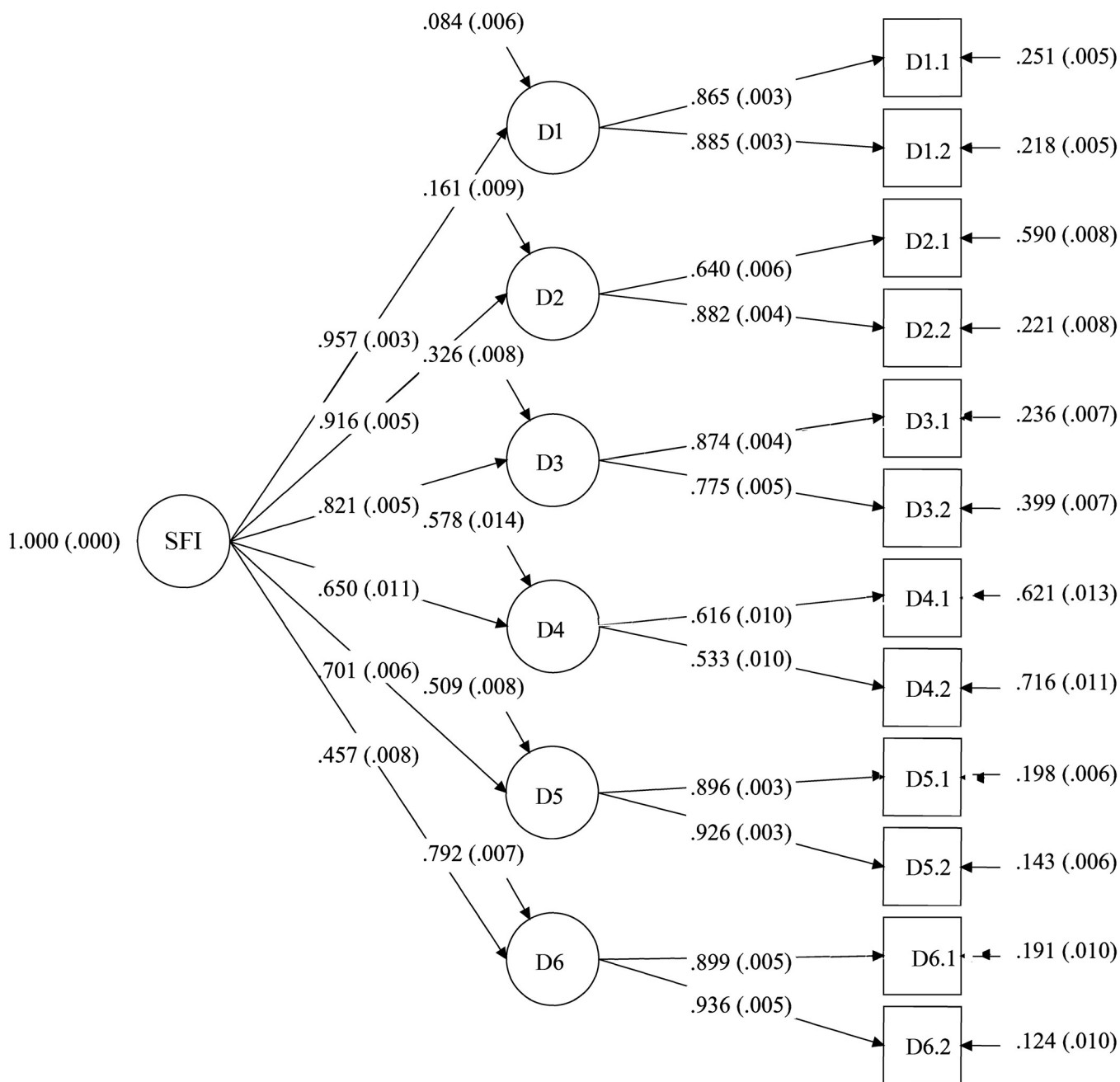

**Fig 3. Standardized factor loadings (with standard errors in brackets) for the second-order confirmatory factor analysis for the Secure Flourish Index (SFI) ($N_{Study\ 2}$ = 13,268).**

## Discussion

As in study 1, few domains had minimally leptokurtic distribution with right-handed tails. Supporting previous results of the *EFA*, the *CFA* suggests that both indexes have hierarchical structures; thus, they consist of items grouped onto their respective domains and the items of the FI and the items of the SFI can be aggregated into composite measures. This is consistent with findings reported by [20].

**Table 6. Fit statistics for the structure of the Flourish Index and Secure Flourish Index ($N_{Study\ 2}$ = 13,268).**

| Model | $\chi^2$ (p-value) | df | CFI | TLI | RMSEA | RMSEA 90% CI | SRMR |
|---|---|---|---|---|---|---|---|
| FI$_{D1-5}$ (five-factor model) | 729.135 (0.000) | 25 | 0.990 | 0.982 | 0.046 | 0.043, 0.049 | 0.019 |
| FI$_{D1-5}$ (second-order factor model) | 2014.192 (0.000) | 30 | 0.972 | 0.959 | 0.071 | 0.068, 0.073 | 0.035 |
| SFI$_{D1-6}$ (six-factor model) | 1163.234 (0.000) | 39 | 0.988 | 0.979 | 0.047 | 0.044, 0.049 | 0.023 |
| SFI$_{D1-6}$ (second-order factor model) | 2478.163 (0.000) | 48 | 0.973 | 0.963 | 0.062 | 0.060, 0.064 | 0.035 |

*Notes*: $\chi^2$ = Chi-Square; *df* = Degrees of Freedom; *CFI* = Comparative Fit Index; *TLI* = Tucker–Lewis Index; *RMSEA* = Root Mean Square Error of Approximation; *CI* = Confidence interval; *SRMR* = Standardized Root Mean Square Residual.

Measurement invariance testing indicates equivalent factor scores and item intercepts for women and men; thus, latent regression coefficients, (co)variances, and mean scores may be compared without bias. For age, results also suggest metric invariance for the SFI and FI and scalar invariance for the SFI; for FI, the intercepts of the items may not be similar for people of different ages. These results provide evidence of the generalizability of the SFI and FI to gender and suggest possible generalizability to age.

Measurement accuracy was again acceptable to excellent for both scales and all sub-domains and with the refined item, it increased also for sub-domain D4 Character and Virtue.

As expected, we found positive correlations of FI and SFI with indicators of social support or self-efficacy, while negative correlations were found with indicators of a fear of failure and diagnosed mental illnesses. While these results show mainly weak to moderate correlations with other relevant but conceptually different measures (suggesting discriminant validity), a test with a direct well-being measure would underline the convergent validity of the scale.

## Study 3

Study 3 aims at testing the convergent validity with an explicit well-being measure. Therefore, this study tests correlations of the Human Flourishing Scale with the German version of the PERMA-Profiler, which is a brief multidimensional measure of psychological well-being [5] in terms of Seligman's PERMA theory [51]. We expect that both measures correlate substantially, while on a domain level, correlations might be substantially lower if a certain domain is not well reflected in each other measure, for example, financial and material stability is hardly reflected in the PERMA-Profiler but in the flourishing scale.

**Table 7. Goodness of fit for gender and age measurement invariance for Flourish Index (FI) and Secure Flourish Index (SFI) models ($N_{Study\ 2}$ = 13,268).**

| Model | FI$_{D1-5}$ (five-factor model) | | | | SFI$_{D1-6}$ (six-factor model) | | | |
|---|---|---|---|---|---|---|---|---|
| | CFI | df | RMSEA | BIC | CFI | df | RMSEA | BIC |
| **Gender invariance** | | | | | | | | |
| Configural | 0.990 | 50 | 0.047 | 513840.355 | 0.987 | 78 | 0.047 | 621088.397 |
| Metric | 0.990 | 55 | 0.045 | 513797.186 | 0.987 | 84 | 0.046 | 621037.940 |
| Scalar | 0.989 | 60 | 0.045 | 513836.910 | 0.986 | 90 | 0.046 | 621071.346 |
| **Age invariance** | | | | | | | | |
| Configural | 0.989 | 75 | 0.048 | 512757.800 | 0.987 | 117 | 0.048 | 619670.078 |
| Metric | 0.988 | 85 | 0.049 | 512801.516 | 0.985 | 129 | 0.048 | 619696.881 |
| Scalar | 0.978 | 95 | 0.061 | 513410.548 | 0.977 | 141 | 0.057 | 620315.603 |

*Notes*: *CFI* = Comparative Fit Index; *df* = Degrees of Freedom; *RMSEA* = Root Mean Square Error of Approximation; *BIC* = Bayesian Information Criterion.

**Table 8. Intercorrelations of the Flourish Index (FI) and Secure Flourish Index (SFI) with validation measures ($N_{Study\ 2}$ = 13,268).**

|  | FI$_{D1-5}$ | SFI$_{D1-6}$ | D1 | D2 | D3 | D4 | D5 | D6 |
|---|---|---|---|---|---|---|---|---|
| Importance of health | 0.191*** | 0.181*** | 0.148*** | 0.124*** | 0.150*** | 0.179*** | 0.150*** | 0.064*** |
| Mental illnesses diagnosed | -0.294*** | -0.298*** | -0.271*** | -0.375*** | -0.212*** | -0.093*** | -0.187*** | -0.167*** |
| Fear of failure | -0.378*** | -0.383*** | -0.320*** | -0.321*** | -0.345*** | -0.225*** | -0.257*** | -0.215*** |
| Berlin Social Support Scale (BSSS) | 0.480*** | 0.467*** | 0.420*** | 0.337*** | 0.348*** | 0.180*** | 0.536*** | 0.202*** |
| General Self-Efficacy Beliefs (ASKU) | 0.396*** | 0.396*** | 0.328*** | 0.339*** | 0.344*** | 0.284*** | 0.256*** | 0.206*** |
| Temper | -0.239*** | -0.241*** | -0.194*** | -0.208*** | -0.202*** | -0.181*** | -0.154*** | -0.131*** |
| Materialism | -0.118*** | -0.133*** | -0.092*** | -0.078*** | -0.088*** | -0.107*** | -0.098*** | -0.118*** |
| Household equivalence income | 0.032*** | 0.039*** | 0.025** | 0.032*** | 0.027** | 0.018* | 0.022** | 0.043*** |
| Possibility to save money | 0.221*** | 0.309*** | 0.226*** | 0.198*** | 0.160*** | 0.139*** | 0.146*** | 0.436*** |

*Notes*: FI = D1-D5; SFI = D1-D6; D1 = Happiness and Life Satisfaction; D2 = Mental and Physical Health; D3 = Meaning and Purpose; D4 = Character and Virtue; D5 = Close Social Relationships; D6 = Financial and Material Stability.

## Methods

**Participants.** For study 3, we recruited adult participants (18 and older) in Germany via the respondi Online Panel, an actively managed panel used for market research with voluntary participation and a double opt-in registration process (in an elaborate scoring and control process, the panel is subjected to permanent quality controlling).We used a nationwide quota sample representative for the sex, age (18–74), and province and, which should allow for a better generalizability of the results as compared to rather selective student samples. In accordance with German data protection regulations, personal data and survey data are stored separately. Of the 388 participating respondents, 345 participants (88.9%) provided written consent. We excluded 28 respondents due to missing data on any of the analyzed variables. Thus, 317 participants (female 49.4%; average age: 45.73 years ranging from 18–74) comprise the analytical sample. Respondents completing the survey received a small incentive (€0.45). Ethics approval was received from the Faculty of Management, Economics and Social Sciences of the University of Cologne (approval numbers: 200028SeSa). The data that support the findings of this study are openly available in the institutional repository of Bielefeld University [31].

**Instruments.** *Flourishing Index (FI) and Secure Flourishing Index (SFI)*. The *FI* and *SFI* were assessed as in Study 1 with item D4.2 being adapted (Table 1, see descriptives in S3 and S4 Tables in S1 File).

*PERMA-Profiler*. We used the German version of the PERMA-Profiler with three items for each of its five dimensions: *Positive Emotions* (P), *Engagement* (E), *Positive Relationships* (R), *Meaning* (M), and *Accomplishment* (A) [5]. As in the original paper, we included eight filler items that aimed at disrupting response tendencies and to provide additional information: one item assessed *overall happiness*, three items assessed *self-perceived physical health*, three items assessed *negative emotions* (sadness, anger, and anxiety), and one item assessed *loneliness*– while *overall happiness* and *perceived physical health* contribute to the understanding of convergent validity, *negative emotions* and *loneliness* add to the discriminant validity testing. Items were rated on 11-point scales ranging from "never" [0] to "always" [10] or "not at all" [0] to "completely" [10]. Items of each dimension were averaged and a total score of the 15 items was computed (see S5 Table in S1 File for descriptives).

**Statistical analysis.** Following other work in this area [e.g., 52], we used Pearson correlations to assess the convergent validity of both scales and their dimensions, whereby higher correlation coefficients suggest indication for stronger convergent validity [53, 54]. Sullivan and

**Table 9. Intercorrelations of the Flourish Index (FI), the Secure Flourish Index (SFI), the six flourishing domains with the PERMA-Profiler, self-perceived health, negative emotions, loneliness, and overall happiness ($N_{Study\ 3}$ = 317).**

| | FI$_{D1-5}$ | SFI$_{D1-6}$ | D1 | D2 | D3 | D4 | D5 | D6 |
|---|---|---|---|---|---|---|---|---|
| Overall well-being | 0.864*** | 0.842*** | 0.759*** | 0.629*** | 0.741*** | 0.536*** | 0.725*** | 0.323*** |
| Accomplishment | 0.703*** | 0.698*** | 0.649*** | 0.595*** | 0.587*** | 0.422*** | 0.518*** | 0.303*** |
| Engagement | 0.388*** | 0.393*** | 0.312*** | 0.275*** | 0.333*** | 0.330*** | 0.295*** | 0.192*** |
| Positive emotions | 0.845*** | 0.816*** | 0.784*** | 0.655*** | 0.715*** | 0.473*** | 0.685*** | 0.294*** |
| Relationships | 0.731*** | 0.706*** | 0.613*** | 0.489*** | 0.534*** | 0.425*** | 0.786*** | 0.256*** |
| Meaning | 0.790*** | 0.759*** | 0.684*** | 0.525*** | 0.792*** | 0.506*** | 0.586*** | 0.262*** |
| Self-perceived health | 0.631*** | 0.631*** | 0.573*** | 0.786*** | 0.450*** | 0.228*** | 0.436*** | 0.288*** |
| Negative emotions | -0.459*** | -0.459*** | -0.409*** | -0.395*** | -0.344*** | -0.302*** | -0.368*** | -0.207*** |
| Loneliness | -0.462*** | -0.433*** | -0.402*** | -0.295*** | -0.342*** | -0.296*** | -0.476*** | -0.118* |
| Overall happiness | 0.774*** | 0.760*** | 0.755*** | 0.556*** | 0.646*** | 0.418*** | 0.653*** | 0.307*** |

*Notes:* FI = D1-D5; SFI = D1-D6; D1 = Happiness and Life Satisfaction; D2 = Mental and Physical Health; D3 = Meaning and Purpose; D4 = Character and Virtue; D5 = Close Social Relationships; D6 = Financial and Material Stability.

*p<0.05

***p<0.001.

Fein [55] provide guidance for classifying the strength of correlative associations (i.e., $r > ±.2$ indicates a small effect, $r > ±.5$ a medium effect, and $r > ±.8$ a large effect).

## Results

Inter-correlations of the Flourish Index (FI, $r = 0.864$) and the Secure Flourish Index (SFI, $r = 0.842$) strongly correlate with the overall well-being score of the PERMA-Profiler (Table 9). While the flourishing domains D1 to D5 substantially correlate with overall well-being, D6 "Financial and Material Stability" that is not directly represented in the PERMA-Profiler correlated less strong ($r = 0.323$). All PERMA-Profiler subscales seem well represented by Flourish Index (FI, $r = 0.703$ to $r = 0.845$) and the Secure Flourish Index (SFI, $r = 0.698$ to $r = 0.816$), the only exception is engagement that seems less represented by the flourishing measure ($r_{FI} = 0.388$ and $r_{SFI} = 0.393$). Moreover, self-perceived health ($r = 0.228$ to $r = 0.786$) and overall happiness ($r = 0.307$ to $r = 0.774$) as further indicators of convergent validity consistently showed positive correlations of small to medium effect sizes with all flourishing measures, while negative emotions ($r = -0.207$ to $r = -0.459$) and loneliness ($r = -0.118$ to $r = -0.476$) as further indicators of discriminant validity showed negative correlations of up to small effect sizes. For more details see Table 9.

## Discussion

We found consistently positive correlations of all PERMA-Profiler measures with the flourishing measures, especially on the scale level (correlations with domains were partially lower). It seems that engagement is less represented in the flourishing measure and financial and material stability is less represented in the PERMA-Profiler instrument. Moreover, the flourishing measures correlated positively with self-perceived health (especially D2 "Mental and Physical Health") and overall happiness (especially FI, SFI, and D1 "Happiness and Life Satisfaction"), which provided evidence for convergent validity, and the flourishing measures correlated negatively with negative emotions (especially FI, SFI, and D1 "Happiness and Life Satisfaction") and loneliness (especially D5 "Close Social Relationships"), providing evidence for discriminant validity.

## Overall discussion

The multidimensional measurement of human flourishing is important to understand whether and through which mechanisms the partially dramatic social, political, and economic changes in societies across the globe affect the human species, in which dimensions, and with which consequences. Such empirical studies are especially relevant for public health, psychology, political sciences, or economics. Especially longitudinal, prospective studies are needed for this, leading to a demand for time and cost-efficient instruments. Therefore, we here provide a translated and validated German version of the twelve-item Human Flourishing Scale that captures complete subjective well-being and the availability of material and financial prerequisites to maintain the state flourishing over time. To this end, we engaged in a multi-stage translation process according to the guidelines for cross-cultural adaptation of health-related measures [28, 29] followed by three validation studies with German-speaking participants to examine the items and scale descriptively, the dimensional structure, measurement invariance across gender and age, internal consistency and the scale's discriminant and convergent validity.

### Descriptive evaluation

Our skewness and kurtosis analysis suggests that the items, sub-dimensions, and scales do not deviated from normality making them suitable for various analyses.

### Dimensional structure

The one- and two-factor structure of the FI and SFI from the EFA is supported by the results of the CFA. The CFA also showed that aggregation at the subscale level is justified. Thus, the items and their respective domains can be aggregated into the two composite measures as theorized [7] and evidenced by findings from previous studies examining the indices across culturally distinct populations [21]. The comparatively lower factor loadings of D4-Character and Virtue D4.2 item are notable but unsurprising given evidence from earlier analysis showing similar patterns [20]. It is possible that the domain of Character and Virtue addresses a broader range of behaviors than the other domains. For example, the two items in our Happiness and Life Satisfaction domain (feeling happy and feeling satisfied) intuitively seem to go together, but as measured by the two items in our Character and Virtue domain, "promoting good" might in some cases might happen quite independent of "delaying happiness." For example, one might find a great deal of happiness in promoting the good. There are no doubt other reasons why D4.2 might have lower factor loadings and given the relative newness of character measures in the flourishing literature—none of the other measures include this domain—this is clearly an area in need of further investigation.

### Internal consistency

Overall, our data indicate acceptable to excellent reliability of the FI and SFI as well as all sub-domains, despite the brevity of the measure. After a slight change in the wording of one item in the domain D4-Character and Virtue, the reliability of the sub-domain increased, but future research may, however, re-evaluate and improve this measure.

### Measurement invariance

Our measurement invariance testing suggests that for FI and SFI the (co)variances latent regression coefficients and mean scores are generalizable for gender and may be compared across women and men without bias. For age, we found metric invariance for the SFI and FI factor structures, which indicates that a one-unit change in either domain has the same

meaning across age groups. We also found scalar invariance for the SFI but not for the FI, which suggests that the intercepts of the items in the FI may not be similar for people of different ages. However, the fit statistics for the more restrictive scalar model still provided sufficient fit suggesting that the latent means may be meaningfully compared across age groups.

## Validity

We investigated construct validity in terms of convergent and discriminant validity by examining correlations with the PERMA-Profiler as well as a range of measures on mental health (e.g., diagnosed mental illnesses), social cohesion (e.g., perceived social support), personal resources (e.g., self-efficacy), and wealth (e.g., household equivalence income). We generally found very strong correlations of both FI and SFI with overall well-being as measure with the PERMA-Profiler providing evidence for high convergent validity. These findings were also evident for all sub-dimensions of the PERMA-Profiler. One exception was the lower correlation with the engagement dimension which seems to be less represented in the flourishing measures. But since the Human Flourishing Scale and the PERMA-Profiler do not fully overlap in their measurement, we expected that substantially lower correlations could occur for the sub-dimensions of the profiler as well as between certain domains of the Human Flourishing Scale. This can be seen, for example, in the weaker correlations of the profiler with financial and material stability, which is hardly reflected in the PERMA-Profiler but in the flourishing scale. Also notably is the strong correlation of the overall happiness measure within PERMA-Profiler with FI and SFI pointing towards high convergent validity. Further support for discriminant validity can be seen in the lower correlations of the Human Flourishing Scale with certain filler items within the PERMA-Profiler such as negative emotions and loneliness (assessed in Study 3) and further measures in Study 2 such as the Berlin Social Support Scale, importance of health, general self-efficacy beliefs, fear of failure, or higher temper indicates.

## Limitations

The present study has some limitations to note. Validation of the scale should include the assessment of the consistency or stability of the translated flourishing measures over time using test-retest reliability, which should be considered in subsequent studies. Additionally, some measures (such as household net income or diagnosed mental illnesses) might be subject to socially desirable responding what could distort our validity analysis. Furthermore, while we focused on the PERMA-Profiler to assess the convergent validity, we encourage future inquiry to also examine the relation of our translated FI and SFI measures with other existing instruments such as the "Fragebogen zur Lebenszufriedenheit" (FLZ) [56] or the German version of the Satisfaction with Life Scale SWLS) [57]. With these limitations in mind, the findings from this study indicate that the German version of the FI and the SFI are appropriate measures for assessing distinct domains of well-being and overall flourishing in the German population.

## Conclusion

In sum, the results of the three studies on the carefully translated and validated German version of the Human Flourishing Scale–consisting of the Flourishing Index (FI) and the Secure Flourishing Index (SFI)–provide support for their use in empirical research to efficiently measure distinct domains of subjective well-being that are universally valued and overall flourishing [4, 7]. These ten, respectively twelve item measures might be suitable for large-scale, multi-themed, and cross-cultural studies to monitor changes and stabilities in well-being and flourishing and for comparisons across countries [20, 21]. Recent research has stressed the value of complementing widely used measures of deficiency, such as loneliness, depression or anxiety,

as well as narrow measures of well-being, such as income, physical health, or life satisfaction alone, with more holistic, positive, and multidimensional measures of complete well-being, or flourishing [4, 7]. The established empirical literature on well-being remains valuable on its own merits, but a more integrated approach offers the possibility of greater insights into more of the valued ends of human life. Measures of flourishing offer assistance in this direction by encompassing a variety of desired domains and inviting a more complete consideration of how these domains complement each other, as well as when during the life course trade-offs might be necessary. Our translation of the FI and SFI, along with the empirical relationships that we found among the measures that we reviewed, will help scholars in Germany (and beyond) explore an expanded range of domains of well-being, including the comparatively neglected domains of character and virtue, physical health, and financial and material stability [6, 58]. Progress in understanding these issues across cultures promises to contribute to a "well-ordered science" of the good life [4].

## Supporting information

**S1 File. Contains all the supporting tables (S1-S5 Tables).**
(DOCX)

## Acknowledgments

We thank those who helped to conduct this study, especially Dana Pietralla, Guido Mehlkop, Fabian Hasselhorn, Floris van Veen, Saskia Huber, and the forsa-team. Moreover, we thank Andrea Steinebrunner, Don MacDonald, Floris van Veen, Frank M. Rosenbauer, Guido Mehlkop, Kevin Phillips, and John Eric Baugher for their contribution to the expert round of the translation of the Secure Flourish Index into German. We also thank Jonas Jakubassa for supporting the data analysis as well as Fiona Seiffert for editorial assistance.

## Author Contributions

**Conceptualization:** Sebastian Sattler, Matthew T. Lee.

**Data curation:** Sebastian Sattler.

**Formal analysis:** Sebastian Sattler, Renae Wilkinson.

**Visualization:** Sebastian Sattler.

**Writing – original draft:** Sebastian Sattler, Renae Wilkinson, Matthew T. Lee.

**Writing – review & editing:** Sebastian Sattler, Renae Wilkinson, Matthew T. Lee.

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
