## [Decision Letter · Decision Letter 0]

1 Mar 2023

PONE-D-22-30338A Brief Measure of Complete Subjective Well-Being in Germany: A Population-Based Validation of a German Version of the Flourish Index (FI) and the Secure Flourish Index (SFI)PLOS ONE

Dear Dr. Sattler,

Thank you for submitting your manuscript to PLOS ONE. After careful consideration, we feel that it has merit but does not fully meet PLOS ONE’s publication criteria as it currently stands. Therefore, we invite you to submit a revised version of the manuscript that addresses the points raised during the review process.

We look forward to receiving your revised manuscript.

Kind regards,

Richard Huan XU

Academic Editor

PLOS ONE

Journal Requirements:

2. Peer review at PLOS ONE is not double-blinded (https://journals.plos.org/plosone/s/editorial-and-peer-review-process). For this reason, authors should include in the revised manuscript all the information removed for blind review.

Reviewers' comments:

Reviewer's Responses to Questions

**Comments to the Author**

1. Is the manuscript technically sound, and do the data support the conclusions?

Reviewer #1: Yes

2. Has the statistical analysis been performed appropriately and rigorously? 

Reviewer #1: Yes

3. Have the authors made all data underlying the findings in their manuscript fully available?

Reviewer #1: Yes

4. Is the manuscript presented in an intelligible fashion and written in standard English?

Reviewer #1: Yes

5. Review Comments to the Author

Reviewer #1: The manuscript aims to develop a German version of FI and SFI in a multi-stage translation and scale testing process on large sample analysis. Several indicators of reliability and validity were used to verify the measurement characteristics of FI and SFI in the German population. This manuscript is very valuable, especially in the field of measurement of subjective well-being. I am very pleased to review this manuscript. I have learned a lot from it. but there are several issues for the authors to consider:

1.Abstract

In general, the end of the abstract should emphasize the significance of the manuscript's conclusions for research in the relevant field, which is obviously lacking in the current manuscript. In addition, the abstract should be brief, so the descriptions about sub-domains of FI and SFI should be removed. Furthermore, It is not recommended to write abbreviations such as "FI" and "FI" directly in the abstract, because abbreviations tend to confuse the reader. If authors insist on abbreviations, it is advisable to use the statement "The Flourish Index (FI) and the Secure Flourish Index (SFI)" when The Flourish Index and the Secure Flourish Index first appear in the abstract.

2.Introduction

It’s OK. However, It would be better to add to the statement about the current state of psychological well-being measurement tools in Germany and the advantages of SI and SFI compared to them.

3.Study 1, 2, 3

a Please add inclusion and exclusion criteria for participants, and add the statement about how to ensure the data quality in Study 1, 2, 3.

b The Instruments section should have a reasonable reference source in Study 1, 2, 3.

c There are some table reference errors in line 244 and line 475. Table 3 seems redundant, please check it. Please check the whole paper for similar errors.

d In study 2, the reliability of validation measures, such as the Importance of health, etc., should be reported.

e In study 3, please explain why Pearson correlation analysis was used to verify the validity of convergence and what are the criteria for using this method. Moreover, in the Discussion, the convergence validity of FI and SFI was not reflected, and the correlation between variables was only simply introduced.

4.Overall discussion

a In the section of Dimensional structure, the comparatively lower factor loading of D4-Character and Virtue D4.2 item should be discussed in more depth from the content of the item, instead of simply comparing the literature.

b Please add limitations and contributions to the literature of the manuscript at the end.

6. PLOS authors have the option to publish the peer review history of their article (what does this mean?). If published, this will include your full peer review and any attached files.

Reviewer #1: No

---

## [Author Response · Author response to Decision Letter 0]

23 Mar 2023

Please see the attached file "Reply to Reviewers".

---

## [Editor Report · Decision Letter 1]

11 Apr 2023

A Brief Measure of Complete Subjective Well-Being in Germany: A Population-Based Validation of a German Version of the Flourish Index (FI) and the Secure Flourish Index (SFI)

PONE-D-22-30338R1

Dear Dr. Sattler,

We’re pleased to inform you that your manuscript has been judged scientifically suitable for publication and will be formally accepted for publication once it meets all outstanding technical requirements.

Kind regards,

Richard Huan XU

Academic Editor

PLOS ONE
---

## [Editor Report · Acceptance letter]

3 May 2023

PONE-D-22-30338R1 

A Brief Measure of Complete Subjective Well-Being in Germany: A Population-Based Validation of a German Version of the Flourish Index (FI) and the Secure Flourish Index (SFI) 

Dear Dr. Sattler:

I'm pleased to inform you that your manuscript has been deemed suitable for publication in PLOS ONE. Congratulations! Your manuscript is now with our production department. 

Kind regards, 

on behalf of

Dr. Richard Huan XU 

Academic Editor

PLOS ONE